# On A. D. Sakharov's Hypothesis of Cosmological Transitions with Changes in the Signature of the Metric [†]

**Tatyana P. Shestakova** 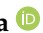

Department of Theoretical and Computational Physics, Southern Federal University,
Sorge St. 5, 344090 Rostov-on-Don, Russia; shestakova@sfedu.ru
† This paper is an extended version from the proceeding paper: Tatyana Shestakova. On A. D. Sakharov's
Hypothesis of Cosmological Transitions with Changes in the Signature of the Metric. In Proceedings of the 1st
Electronic Conference on Universe, online, 22–28 February 2021.

**Abstract:** The paper discusses possible consequences of A. D. Sakharov's hypothesis of cosmological transitions with changes in the signature of the metric, based on the path integral approach. This hypothesis raises a number of mathematical and philosophical questions. Mathematical questions concern the definition of the path integral to include integration over spacetime regions with different signatures of the metric. One possible way to describe the changes in the signature is to admit time and space coordinates to be purely imaginary. It may look like a generalization of what we have in the case of pseudo-Riemannian manifolds with a non-trivial topology. The signature in these regions can be fixed by special gauge conditions on components of the metric tensor. The problem is what boundary conditions should be imposed on the boundaries of these regions and how they should be taken into account in the definition of the path integral. The philosophical question is what distinguishes the time coordinate among other coordinates but the sign of the corresponding principal value of the metric tensor. In particular, there is an attempt in speculating how the existence of the regions with different signature can affect the evolution of the Universe.

**Keywords:** A. D. Sakharov; metric signature; path integral quantization; unitary evolution

## 1. Introduction

In 1979, S.W. Hawking emphasized the importance of taking into account various spacetime topologies in quantum gravity [1]:

> "...one would expect that quantum gravity would allow all possible topologies of spacetime... It is precisely these other topologies that seem to give the most interesting effects."

Andrei Sakharov has made a more radical conjecture. In his paper "Cosmological transitions with changes in the signature of the metric" [2] published in 1984, he wrote:

> "It is conjectured that there exist states of the physical continuum which include regions with different signatures of the metric..."

In particular, the regions may be purely spatial. Then,

> "Differences in the signature structure... appear just as natural as differences in the topological structure."

It was written 37 years ago. I should say, now the conjecture is as difficult to verify as it was 37 years ago. I shall try to discuss some of its consequences, mathematical as well as philosophical. Some questions are: How does one define the path integral over a purely spatial region? What distinguishes the time coordinate among other coordinates? How does the existence of regions with different signatures affect the evolution of the Universe?

In this paper, I shall try to show how ideas of Sakharov inspire reflections on what a future quantum theory of gravity must be. Therefore, though his ideas may seem to be too

fantastic and far from physical reality, they are closely related with the most fundamental questions of modern science.

I shall start from the mathematical points, let us assume that the physical continuum includes regions $U_1, U_2, \ldots$, with the signature $(-, +, +, +)$ and regions $P_1, P_2, \ldots$, with the signature $(+, +, +, +)$. Sakharov made this choice of the signs of principal values of metric tensor following the tradition of Einstein, Minkowski, Landau, and Lifshitz, etc. The notations U and P were introduced by Sakharov and originated from the word "universe" and the name of Parmenides, the Greek philosopher who had argued for a world without motion.

It is worth emphasizing that P regions are regions without time rather than without motion. There are several well-known models of a static universe. Let me remind the first cosmological model proposed by Einstein in 1917, soon after his formulation of general relativity, where he introduced a cosmological constant [3], and the steady-state theory of the expanding Universe of Hoyle, Bondi, and Gold [4,5]. However, the existence of time is assumed in these models, so that the observer could detect very small changes at large scales. In addition, it is well known that a gravitational field created by more than one body cannot be constant; gravitational interaction causes bodies to move that implies the existence of time. However there is no time within purely spatial regions discussed in the context of Sakharov's paper, and the observer cannot exist inside them.

Is it reasonable to assume that P regions exist in our Universe? In 1936, Matvei Bronstein pointed out [6] that at the Planck scale a spacetime structure cannot be determined since any attempt to do it would disturb this structure. Therefore, one can assume that at the Planck scale, spacetime has an arbitrary topological structure. Moreover, one can go beyond and suppose that the existence of regions with various signatures is also possible. Here is an additional argument in favor of Sakharov's hypothesis. However observational effects of the regions of these scales would be negligible and could hardly be detected by modern instruments. One can expect that the existence of P regions of a larger size is less probable the larger the size.

In general, the problem of signature changing is much broader than the consideration of timeless P regions inside a universe like ours. In the literature, authors mainly discuss transitions from a P region to an U region (see, for example, [7–11]), that was inspired by attempts to describe the quantum creation of the Universe as such a transition and originated from the works by Vilenkin [12] and Hartle and Hawking [13], which Sakharov was acquainted with, as he mentioned in [2]. In the cited papers, the signature change implies that a temporal coordinate $x_0$ becomes a spatial one, or vice versa. It can be formally reached if one admits complex-valued transformations like $t \rightarrow -iy$, that causes a $g_{00}$-component of the metric tensor to change its sign. In this case, coordinate transformations are extended to include complex-valued ones, but only those that touch the $x_0$ coordinate. The class of admissible transformations can be extended further to involve coordinates usually considered as spatial. It would enable one, for example, to explore a situation when a temporal coordinate becomes spatial while a spatial coordinate becomes temporal. Though this possibility may be intriguing from a purely speculative point of view, there is no strong motivation for it.

Therefore, in this paper I shall confine myself to the consideration of transformations that touch only one coordinate $x_0$. In [2], Sakharov supposed to use path integrals, but he has not given a strict definition of the path integral ignoring gauge fixing and other problems. In the next section, I shall consider a mathematical definition of the path integral over a purely spatial region. In the class of transformations discussed above, the signature in different regions can be fixed by special gauge conditions. It leads to the question if the requirement of gauge invariance of the path integral is applicable in this case. The problem of gauge invariance in the case of spacetime with non-trivial topology has been already analyzed in our papers [14–17] in which the so-called extended phase space approach to quantization of gravity was proposed. In Section 3, the case when the Universe has a non-trivial topology is briefly outlined from the viewpoint of this approach. It will be

compared with the situation when the physical continuum includes regions with different signatures of the metric tensor.

If one defines the path integral as a result of the replacement $t \to -iy$ (that is confirmed by the example suggested by Sakharov), one would come to the conclusion that, in a P region, the path integral could be treated as a matrix element of a non-unitary operator. Its effect on the evolution of the Universe is discussed in Section 4. In accordance with Sakharov's hypothesis, a possible existence of regions with additional time dimensions is also considered. The final discussion and conclusions are given in Section 5.

## 2. The Definition of the Path Integral over Spatial Regions

The path integral over the physical continuum and some matter distribution should have the form:

$$\int \prod_{x \in U_1} \left\{ \prod_{\mu,\nu} dg_{\mu\nu}(x) \, d\phi(x) \, M\big[g_{\mu\nu}(x), \phi(x), U_1\big] \exp(iS[U_1]) \right\} \times$$

$$\times \prod_{x' \in P_1} \left\{ \prod_{\mu,\nu} dg_{\mu\nu}(x') \, d\phi(x') \, M\big[g_{\mu\nu}(x'), \phi(x'), P_1\big] \exp(iS[P_1]) \right\} \dots \tag{1}$$

where $\phi(x)$ stands for matter fields. As we can see, the path integral is factorized into a product of integrals over the regions $U_1$, $U_2$, …, and regions $P_1$, $P_2$, …; $M\big[g_{\mu\nu}, \phi\big]$ is a measure in the path integral which can be different in different regions.

Let us consider a physical continuum which consists of three regions $U_1$, P, and $U_2$ (see Figure 1a). The region P has spacelike boundaries $B_1$ and $B_2$. In the U regions the path integral is a probability amplitude $\langle g_2, \phi_2, S_2 | g_1, \phi_1, S_1 \rangle$ to go from a state with a spacetime metric $g_1$ and matter fields $\phi_1$ on a hypersurface $S_1$ to a state with a spacetime metric $g_2$ and matter fields $\phi_2$ on a hypersurface $S_2$. It is a sum over all field configurations $g$ and $\phi$ [1,13,18]. If hypersurfaces $S_1$, $S_2$ correspond to some points in time, $t_1$ and $t_2$, one can formally consider the probability amplitude as a matrix element of the evolution operator:

$$\langle g_2, \phi_2 | \exp\big[-i\hat{H}(t_2 - t_1)\big] | g_1, \phi_1 \rangle. \tag{2}$$

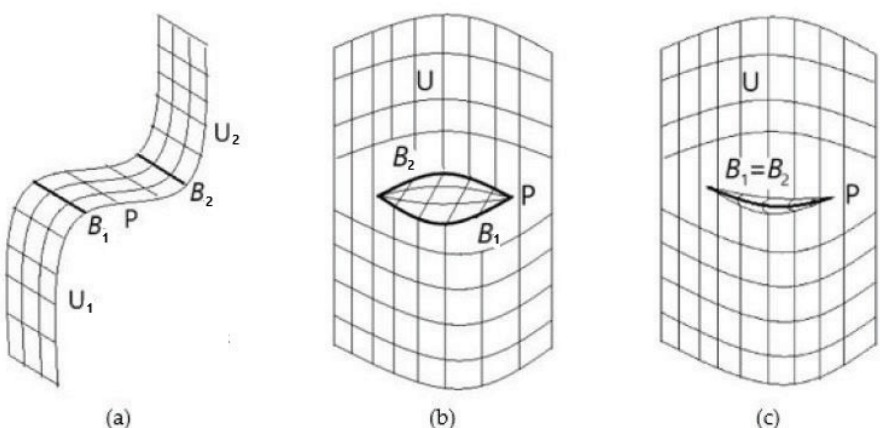

(a)  (b)  (c)

**Figure 1.** Examples of physical continuums with regions with various signatures.

To describe the changes in the signature of the metric, one can admit that time (as well as space) coordinates may be purely imaginary. It is the Wick rotation technique that is often used in quantum field theory and quantum gravity, though it can be considered just as a formal mathematical trick.

Sakharov gave the example of the change in the signature that may seem somewhat artificial. Let the component $g_{00}$ of the metric tensor change its sign at the boundaries $B_1$ and $B_2$. This component depends on a time coordinate $x_0$ such as:

$$g_{00} = \frac{l(x)}{x_0 - a}. \tag{3}$$

Sakharov supposed that, in general, $l(x)$ may be a function of spatial coordinates, but it does not change in order of magnitude. This supposition takes us beyond the class of transformations discussed above, so we shall assume below that $l$ is a constant or its derivatives are negligible.

We can see that $g_{00}$ is singular at the point $x_0 = a$ at the boundary. Under the assumption about $l$, using the formula for covariant tensor transformation,

$$g'_{\mu\nu} = \frac{\partial x^\lambda}{\partial x'^\mu} \frac{\partial x^\rho}{\partial x'^\nu} g_{\lambda\rho}, \tag{4}$$

it is easy to check that the singularity of $g_{00}$ can be eliminated by the following coordinate transformation:

$$x_0 - a = \frac{y^2}{4l}, \quad g_{00} \to g'_{00} = 1 \quad \text{(in the P region)};$$

$$a - x_0 = \frac{t^2}{4l}, \quad g_{00} \to g'_{00} = -1 \quad \text{(in the U region)}, \tag{5}$$

(see Equations (2) and (3) in [2]). Therefore, the transition across the boundary corresponds to the replacement $t \to -iy$ that resembles the Wick rotation. Then, the path integral over the region P can be presented as:

$$\langle g_2, \phi_2 | \exp\left[-\hat{H}(y_2 - y_1)\right] | g_1, \phi_1 \rangle. \tag{6}$$

The boundaries $B_1$ and $B_2$ play the role of hypersurfaces $S_1$ and $S_2$. It is a matrix element of a non-unitary operator, meanwhile the values of the fields do not necessarily match at the boundaries $B_1$ and $B_2$, in other words, the states $| g_1, \phi_1 \rangle$ and $| g_2, \phi_2 \rangle$ may be different.

We have seen that the assumption of the regions with various signatures implies introducing different coordinate sets in different regions of the physical continuum. One meets a similar situation in the case of pseudo-Riemannian manifolds with a non-trivial topology, when one coordinate set is not enough to describe the geometry of the manifold. Thus, the consideration of a physical continuum like the one described above looks as a further generalization.

It is worth noting that one can find solutions to the Einstein equations for a metric with the "Lorentzian" signature $(-, +, +, +)$ as well as for a metric with the "Euclidean" signature $(+, +, +, +)$. It has been demonstrated, for example, in the work by Ellis et al. [7], where the obtained classical solutions are analogues of what is implied by the Hartle–Hawking "no boundary" proposal [13], namely, that the Universe is believed to appear from a P region which is a half of 4-sphere, so that the physical continuum is singularity-free. However, the classical solutions in P and U regions should be joined to be continuous on the boundary of these regions. In the approach proposed in [7], the $g_{00}$-component of the metric tensor changes its sign on the boundary, but the full solution turns out to be discontinuous. Ellis et al. argued that the $g_{00}$-component is a non-physical variable which can be chosen arbitrarily, and its discontinuity does not seem to be a big trouble. This point was strongly criticized by Hayward [8].

On the topic of the quantum description of the Universe, the path integral approach implies that the wave function of the Universe satisfies to some Schrödinger equation. The standard procedure of derivation of the Schrödinger equation from the path integral [19,20] involves an approximation of the action using classical solutions. For the gravitational theory, as well as for other gauge theories, the action of the original theory is known to be replaced by an effective action, the latter including a gauge-fixing term. As a rule, gauge conditions aim at fixing some reference frame; on the other hand, they restrict admissible

coordinate transformations. For example, imposing the conditions $g_{00} = -1$ in the U region and $g_{00} = 1$ in the P region obviates the inverse transformation with respect to (5):

$$\begin{cases} y = 2\sqrt{l(x_0 - a)} & \text{(in the P region)} \\ t = 2\sqrt{l(a - x_0)} & \text{(in the U region)} \end{cases} \qquad g_{00} \to g'_{00} = \frac{l}{x_0 - a}, \qquad (7)$$

since the transformation breaks down the conditions for $g_{00}$. These conditions also prohibit complex-valued transformations like $t \to -iy$.

Hence, the signature in different regions of the continuum can be fixed by special gauge conditions on components of the metric tensor. It may be the mentioned above conditions, otherwise some condition may be imposed on the determinant of the metric tensor (it is interesting to note that Weinberg [21] suggested to put restrictions on the determinant, but it aimed at obtaining the Einstein equation with a cosmological constant).

It is generally accepted that quantum theory of gravity must be gauge invariant. In particular, the path integral giving the probability amplitude between two physical states must be gauge invariant, and the Schrödinger equation derived by means of the standard procedure from the path integral must not depend on the chosen gauge conditions. However, can we require gauge invariance of the path integral if gauge conditions fix the signature of the metric? Can the Schrödinger equation be insensitive to the signature change and the $g_{00}$ discontinuity? These requirements would mean that the signature has no observable effect. I would rather expect that the Schrödinger equation should have different forms in P and U regions, so that we should, in essence, deal with different equations for the wave function of the Universe and face the problem of joining their solutions similar to the junction of classical solutions to the Einstein equations in "Euclidian" and "Lorentzian" regions. In the classical case, one can avoid the discontinuity at the boundary of the P and U regions, if one could find a coordinate chart covering the surface of signature changing which agrees with charts on both sides of the boundary. In the quantum case, the problem can be solved if the form of the Schrödinger equation depends smoothly on coordinates in the region covering the surface of the signature change, then one could expect the solutions to be continuous. Anyway, it is a non-trivial mathematical problem by itself.

## 3. Non-Trivial Topology and Gauge Invariance

In the previous section, we mentioned a possibility that the signature may be fixed by means of some conditions on components of the metric tensor that leads to the problem of gauge invariance. As mentioned, Sakharov ignored the problem of gauge choice in the definition of the path integral. However, I believe that it deserves attention. The problem of gauge invariance has been discussed in the framework of the extended phase space approach [14–17] keeping in mind non-trivial topology of spacetime.

In modern quantum theory of gauge fields, the effective action has the following structure:

$$S_{(eff)} = S_{(grav)} + S_{(gf)} + S_{(ghost)}, \qquad (8)$$

where the gauge-fixing term $S_{(gf)}$ and the ghost term $S_{(ghost)}$ are not gauge-invariant. Gauge invariance can be restored by imposing asymptotic boundary conditions on Lagrange multipliers of gauge conditions and ghosts. Physically, the boundary conditions correspond to asymptotic states, which are usually presumed in laboratory experiments, when one has a goal to obtain probabilities of various processes, as in collider physics. The asymptotic boundary conditions play an important role in the Faddeev–Popov approach [22] and in the Batalin–Fradkin–Vilkovisky approach [23–25] to the quantization of gauge theories, but they are justified for a system with asymptotic states. The methods work correctly in the $S$-matrix theory (for which they were elaborated originally), but many authors apply them without a doubt to cosmological models (see, for example, [26]). In gravitational theory, however, with the exception of the special case of asymptotically flat

spacetime, one meets another situation, a gravitating system does not posses asymptotic states and the boundary conditions are not justified.

What result would we come to if we reject imposing the asymptotic boundary conditions? The answer to this question has been obtained in the extended phase space approach to the quantization of gravity [14–17]. The equation for the wave function of the Universe appears to depend on gauge conditions. The Wheeler–DeWitt equation loses its sense as an equation which expresses gauge invariance of the theory, but in any case the wave function must satisfy the Schrödinger equation [27]. For a model with a finite number of degrees of freedom, the general solution to the Schrödinger equation has the following structure:

$$\Psi(N, q, \theta, \bar{\theta}; t) = \int \Psi_k(q, t)\, \delta(N - f(q) - k)\, (\bar{\theta} + i\theta)\, dk. \tag{9}$$

Here $N$ is the only gauge degree of freedom in this model (the lapse function) that is subject to the condition $N = f(q) + k$, $q$ stands for physical degrees of freedom, $k$ is some constant, $\theta$, $\bar{\theta}$ are ghost fields. The wave function (9) contains information about the geometry of the model as well as about the gauge condition which characterizes the state of the observer in accordance with the spirit of general relativity. The function $\Psi_k(q, t)$ can be called a physical part of the wave function, it satisfies a Schrödinger equation with a Hamiltonian operator $\hat{H}_{(phys)}$ depending on the gauge condition.

As a result, the spectrum and eigenfunctions of the operator $\hat{H}_{(phys)}$ will depend on a chosen gauge condition. The measure in the physical subspace will also depend on the gauge-fixing function $f(q)$, as it follows from the normalization condition for the wave function (9):

$$\begin{aligned}
&\int \Psi^*(N, q, \theta, \bar{\theta}; t)\, \Psi(N, q, \theta, \bar{\theta}; t)\, M(N, q)\, dN\, d\theta\, d\bar{\theta} \prod_a dq^a \\
=\ &\int \Psi_k^*(q, t)\, \Psi_{k'}(q, t)\, \delta(N - f(q) - k)\, \delta(N - f(q) - k')\, M(N, q)\, dk\, dk'\, dN \prod_a dq^a \\
=\ &\int \Psi_k^*(q, t)\, \Psi_k(q, t)\, M(f(q) + k, q)\, dk \prod_a dq^a = 1.
\end{aligned} \tag{10}$$

Therefore, in a large degree the whole structure of the physical Hilbert space is determined by the chosen gauge condition (the reference frame). One cannot give a consistent quantum description of the Universe without fixing a certain reference frame, as well as one cannot find a solution to the classical Einstein equations without imposing some gauge conditions.

In the case of non-trivial spacetime topology, one should admit spacetime manifolds with horizons and other peculiarities, so that the metric tensor may be singular at some points. In this case, one needs to introduce more than one coordinate sets (reference frames) to describe the geometry of such a manifold.

Consider a spacetime manifold consisting of several regions $R_1$, $R_2$, …, in each of them varying gauge conditions $C_1$, $C_2$, …, being imposed [28]. It is natural to think that such regions exist in a universe with a non-trivial topology. Just for simplicity, one can assume that boundaries $S_1$, $S_2$, …, between the regions are spacelike and can be labeled by some time variables $t_1$, $t_2$, ….

Within the region $R_1$, the evolution of the physical subsystem is determined by an unitary operator $\exp\left[-i\hat{H}_{(phys)1}(t_1 - t_0)\right]$, where $\hat{H}_{(phys)1}$ is a physical Hamiltonian in the region $R_1$ with gauge conditions $C_1$. At the instant $t_0$ on the hypersurface $S_0$ the state of the system is given by some vector $|g_0, \phi_0\rangle$. Then the state on the boundary $S_1$ reads:

$$|g_1, \phi_1\rangle = \exp\left[-i\hat{H}_{(phys)1}(t_1 - t_0)\right]|g_0, \phi_0\rangle. \tag{11}$$

However, if we go from the region $R_1$ to $R_2$, we would find ourselves in another Hilbert space with a basis formed from eigenfunctions of the operator $\hat{H}_{(phys)2}$. The transition to a new basis is not an unitary operation, as follows from the dependence of the measure in the physical subspace (10) on gauge conditions. Denote the operator of the transition to a new basis in the region $R_2$ as $\hat{P}(t_1)$. Then the initial state in the region $R_2$ is:

$$\hat{P}(t_1)\exp\left[-i\hat{H}_{(phys)1}(t_1 - t_0)\right]|g_0, \phi_0\rangle \tag{12}$$

and

$$
\begin{aligned}
|g_3, \phi_3\rangle &= \exp\left[-i\hat{H}_{(phys)3}(t_3 - t_2)\right]\hat{P}(t_2)\exp\left[-i\hat{H}_{(phys)2}(t_2 - t_1)\right] \times \\
&\times \hat{P}(t_1)\exp\left[-i\hat{H}_{(phys)1}(t_1 - t_0)\right]|g_0, \phi_0\rangle.
\end{aligned}
\tag{13}
$$

We have come to the conclusion that at any border $S_i$ between regions with different gauge conditions unitary evolution is broken down. The operators $\hat{P}(t_i)$ play the role of projection operators, which project states obtained by unitary evolution in a region $R_i$ on a basis in Hilbert space in neighbor region $R_{i+1}$.

In general, time evolution can be presented by the sequence of operators:

$$
\hat{U}(t_N, t_{N-1})\hat{P}(t_{N-1})\hat{U}(t_{N-1}, t_{N-2})\dots\hat{U}(t_3, t_2)\hat{P}(t_2)\hat{U}(t_2, t_1)\hat{P}(t_1)\hat{U}(t_1, t_0).
\tag{14}
$$

In this simple example we have seen that periods of evolution alternate with abrupt changes described by the operators $\hat{P}(t_i)$. We can encounter something similar if we adopt the hypothesis about the existence of regions with different signatures of the metric. In this case, we also deal with different coordinates and different gauge conditions in distinct regions, and the metric tensor can be singular at the boundaries of the regions. Thus, already at the formal mathematical level the hypothesis of Sakharov makes us look at the problem of gauge invariance of quantum gravity from a new point of view.

## 4. Time Coordinates and Evolution in Time

It is well known that there are two meanings of time in physics. Time can be considered as just one of the coordinates that is distinguished from the others by the sign of the corresponding principal value of the metric tensor. However time is also a parameter of evolution of a physical system. Most physical theories ignore time asymmetry. Sakharov admitted cosmological models of the Universe with reversal of time's arrow. In [29], he considered the point of minimal entropy. Since entropy increases in both directions from this point, and the arrow of time is defined by the increase in entropy, time must reverse at this point. The reversal of time's arrow can also occur at the moment of maximal expansion. It is possible in models with infinite repetition of cycles of expansion and contraction as well (in pulsating, or "many-sheeted" models, according to Sakharov's terminology). It seems that he did not suppose that time irreversibility is an inherent property of the Universe. However, Sakharov has not put forward a hypothesis about time reversal in separate regions of our Universe. Returning to the main subject of this paper, we can pose the question: How can the existence of the regions with a different signature affect the evolution of the Universe?

According to a generally accepted point of view, the changes of physical states in the U regions are described by the matrix element (2) of an unitary operator (and the notation U may originate not only from the word "universe", but as well from the word "unitary").

One can imagine the region P inside of the region U as depicted in Figure 1b. The observer living in the region U can reveal an instantaneous change of the fields at the moment $t_0$ corresponding to the location of the region P in time. In accordance with the supposition of Section 2, the temporal coordinate $x_0$ must become a spatial one when crossing the boundary of P. Of course, the fields can vary continuously with this coordinate. However, if the observer is able to measure the values of the fields at the moments $t_0 - \varepsilon$ and $t_0 + \varepsilon$, while $\varepsilon \to 0$, they would detect a "jump" in these values, so that it would be perceived by the observer, who cannot exist in the region P, as a discontinuity of the fields. Therefore, the existence of P regions may lead to additional quantum uncertainty, because of the difference in the states at the boundaries $B_1$ and $B_2$ referring to the same time point.

Theoretically, one can imagine that the boundaries $B_1$ and $B_2$ are glued together, as depicted in Figure 1c, if there exists a one-to-one mapping of the boundary $B_1$ into $B_2$. Then, the region P is compactified. As in the situation in Figure 1b, the fields may vary

within this region. Again, the observer can fix some "jump" in the values of the fields. Thus, the observer existing in time in the region U can hardly feel any difference between the situations shown in Figure 1b,c. In compliance with (6), the state at the boundary $B_1$ can be considered as a result of acting of a non-unitary operator that can be denoted as $\hat{P}(t)$, which corresponds to the notation of the regions.

Let us assume that there exists a number of P regions in our Universe located at time moments $t_1, t_2, \ldots, t_{N-1}$. Again, time evolution can be presented by the sequence of operators (14), but with $\hat{P}(t)$ operators having another sense.

This situation, of course, differs from the situation considered in Section 3. However, in both cases one can say that a non-trivial topological structure (in a broad sense) can be a cause of a possible violation of an unitary evolution.

In his paper [2], Sakharov wrote about another situation, when U regions (may be, even an infinite number of U regions) exist inside the P region, and the creation of our Universe, as well as many other universes, can be thought as a result of quantum transitions with a change in the signature of the metric. Sakharov pointed out that this idea had been inspired by the paper of Vilenkin [12]. In my opinion, this idea requires further elaboration since it poses a difficult question about the appearance of time itself from a timeless continuum.

We can return to the idea of Bronstein that a spacetime structure cannot be detected at the Planck scale. It means that spacetimes of all imaginable topologies and signatures may be equally presented at this scale. Therefore, one can assume that in the very beginning the Universe may be in a state that encompasses equally all possible geometries and gauge conditions. This state can be described by a path integral averaged over a variety of gauge conditions and, in a sense, it can be thought of as gauge-invariant. However it is extremely difficult to elaborate a mechanism explaining how our Universe has been stood out so that one can speak about the appearance of macroscopic time. Formal coordinate transformations like $y \rightarrow it$ can hardly help; they could say nothing about the phenomenon of the creation of the Universe. An explanation that one can find in the literature suggests validity of a semiclassical approximation to quantum gravity that leads to an approximate Schrödinger equation of non-gravitational fields with respect to the semiclassical background (see, for example, [30]). However making use of the semiclassical approximation implies that a classical spacetime has already come into being [31].

It seems that at least two conditions should be satisfied for some coordinate, regarded as a macroscopic temporal coordinate. Some etalon process to measure time as well as some irreversible process to point out a time arrow must take place. At present one can hardly judge what processes may serve as time etalons at a quantum gravitational level. However, in the Very Early Universe, the existence of essentially nonstationary processes is assumed (in this connection, see the paper of Sakharov devoted to baryon asymmetry of the Universe [32]). The principal problem is to describe the appearance of time itself as an evolution parameter.

Following many authors, Sakharov emphasized that the number of dimensions of observed space, being equal to 3, is selected by the fact that atoms and planetary systems would be unstable for a different number of dimensions. It was noted yet by Ehrenfest [33] in 1917. Similar ideas were developed by Dicke, Idlis, and others (see the references in [2]). In 1984, when Sakharov's work was published, Gott and Alpert [34] showed that no gravitational attraction between masses exists in (2 + 1) spacetime. Therefore, any additional dimensions, be they temporal or spatial, must be compactified.

Sakharov suggested that, except for the observable macroscopic time dimension, an even number of additional time dimensions may exist in our Universe. These dimensions are assumed to be compactified. Sakharov believed that the property of global ordering with respect to the macroscopic time is preserved for any signatures. He supposed that the existence of additional time dimensions does not affect macroscopic processes with the participation of particles with energies much less than the radius of the time compactification. We have the next question: What distinguishes these compactified time dimensions from compactified spatial dimensions? It seems that the very notion of time implies

some changes of fields with time coordinates even if the compactification radius is small enough. Since the external observer cannot measure the values of any time coordinates but macroscopic time only, the observer could judge about the existence of compactified time dimensions by sudden changes of the values of the fields (if the instruments are sensitive to the changes), similar to what we have in the case of the existence of (compactified) P regions. The path integral over the regions with additional time dimensions is likely to have the form (2), so the evolution with respect to the time coordinates would be described by an unitary operator. It may be the only difference with purely spatial regions.

## 5. Discussion

In a series of papers on gravitation and cosmology, Andrei Sakharov put forward a number of hypotheses which look exotic even for this purely theoretic field of study. In this context, by the word "exotic" I mean "not experimentally verifiable". At present, as well as several decades ago, when he wrote his papers, it is still difficult to find evidences in favor of or against his hypotheses. We can try to assess the hypotheses by just relying upon mathematical construction such as the path integral.

It is well known that Albert Einstein based upon firm empirical facts in the beginning of his research activity, but later came to rely on mathematics. He wrote [35]:

*"It is my conviction that pure mathematical construction enables us to discover the concepts and the laws connecting them which give us the key to the understanding of the phenomena of Nature."*

His contemporaries believed that he had chosen an incorrect way, which could not have led to significant results. Nevertheless, it would be also wrong to deny any attempts to anticipate a possible development of the theory.

To my mind, the hypothesis about the regions with various signatures of the metric is a chance to think how the physical continuum with a complicated topological structure must be described. Seemingly, one cannot avoid introducing different coordinates in different regions of the continuum. Can it be consistent with the requirement of the theory's gauge invariance? While this requirement is well grounded and is shared by most physicists, the situation with quantum gravity may be different. Nevertheless, many physicists believe that the time evolution must be unitary, even though it contradicts to the irreversibility of observed physical phenomena. A good example is the attempt to resolve the black hole information paradox [36] by means of the AdS/CFT correspondence, which implies that a black hole can evolve in a manner consistent with quantum field theory, and this type of evolution is believed to be unitary.

However, some scientists think otherwise. In many of his papers and books, Roger Penrose advocates the idea that an understanding of irreversibility of physical processes may be closely related with the progress in constructing the quantum theory of gravity [37,38]. Ilya Prigogine also thought that symmetry in time quantum dynamics described by the Schrödinger equation should be generalized to involve irreversible processes. To do it, one has to extend the class of admissible quantum operators beyond Hermitian operators and include non-unitary transformations of state vectors or density matrices [39,40]. However, it would not be appropriate to introduce "by hands" some special interaction which would result in the breakdown of unitarity. On the contrary, it would be desirable if the breakdown of unitary evolution of a physical system follows in a natural way from the very structure of the theory. In the present paper, we have seen some indications that a future quantum theory of gravity may include this potential.

Thanks to the exciting ideas of Sakharov, we have an opportunity to discuss these intriguing problems from a new perspective. I hope that it can give us if not a key, then a hint as to how to reach a deeper understanding of the phenomena of nature.

**Funding:** This research received no external funding.

**Acknowledgments:** I am grateful to M. Yu. Khlopov for invitation to participate in the Session "The Universe of Andrei Sakharov".

**Conflicts of Interest:** The author declares no conflict of interest.

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
