# Peer review of "On A. D. Sakharov’s Hypothesis of Cosmological Transitions with Changes in the Signature of the Metric"

_universe, doi:10.3390/universe7050151_

Round 1

Reviewer 1 Report

This paper draws attention to an old and speculative idea of Sakharov´s, and comments on some mathematical and conceptual aspects of it. Sakharov´s idea and the author´s comments are interesting, but the paper´s discussion is very brief (5 pages) and inconclusive. It indicates some possible questions rather than to engage in a real analysis. As a brief report on an exploratory conference talk it may well be publishable, but I would recommend that the authors adds clarifying remarks here and there in order to make their comments easier to understand. In addition, there are some minor linguistic oversights that can easily be corrected. I have indicated most of them in the pdf.

Some points on which I felt the need for clarification are the following.

In formula 1 phi and K are not defined (although phi is defined later in the context of another formula). It would surely help if some words were added about the meaning and intended role of expression 1. K returns later in the paper, seemingly as a time, but without any clear function.

In formulas 3 and 4 the quantity l is not defined. I did not see immediately how the result of 4 follows.

Six lines below equation 5 it is concluded that seen from region U there will be a change that is *discontinuous*.  That was not immediately clear to me, since it seems that region P may contain a field that varies continuously in *space*, even though all of P´s positions occur at the same time as judged from U. Relatedly, it was not obvious to me that gluing the borders of P together (p. 4) does not change the situation. Some more explanation on these points would be helpful.

Still  on page 3, I found the (very general) remarks about gauge confusing. In what sense could a change of signature be a mere change of gauge? Some more specific argumentation would help.

Author Response

Dear Reviewer,

Please find my response in the attached file.

Sincerely yours,
Tatyana Shestakova.

Reviewer 2 Report

This paper presents a brief extension of an original idea of Sakharov's about
the possible contribution of Euclidean regions to the path integral of Lorentzian quantum gravity. In my opinion, the paper is interesting and should be published, but I ask that the following issues be clarified:

The notation is not very clear in equation (1). There is only one integral sign which suggests that the product of exp(iS[U]) and exp(iS[P]) is being integrated, but the appearance of two identical dg_{mu nu}(x) and dphi(x) would rather indicate the multiplication of two integrals, one of exp(iS[U]) and one of exp(iS[P]). The meaning should be made more clear using parentheses and additional integral signs as well as independent integration variables or positions x for the two kinds of regions. Also, I do not see the relevance of K at the end of the equation.

Equation (3) is a very special example of a signature-changing metric that does not appear to be generic. In particular, the author shows how the apparent singularity can be removed, but in general classical signature change has been found to be singular, see for instance Class Quantum Grav 9 (1992) 1535, gr-qc/9303034 or gr-qc/9610063. It is not clear to me whether this property is a problem for the author's approach, but it should be stated.

I do not understand the discussion of gauge invariance in the context of space-time signature on page 3. The gauge invariance relevant for gravity is given by coordinate transformations, and coordinate transformations do not change the signature of the metric. How is it then possible that "the signature in different regions of the continuum can be fixed by special gauge conditions on components of the metric tensor"? In the example provided, if one chooses g_{00}=-1 as a gauge condition, the remaining metric components would have to be such that det g<0 in Lorentzian signature and det g>0 in Euclidean signature. Thus, x^0 may longer be a time coordinate depending on the gauge fixing, but a different coordinate will be time if the signature is Lorentzian. However, a gauge condition does not determine or change the signature.

On page 4, the statement "In a general case, in the theory of gravity we do not have asymptotically flat spacetime. One should admit spacetime manifolds with horizons and other peculiarities" is unclear because horizons may well exist in asymptotically flat space-times. 

Author Response

(The authors gave the same response as above.)

Reviewer 3 Report

The purpose of this talk is to review Sakharov's ideas of 1984 from the modern perspective. In this respect, I have the following proposals aimed at better fulfilling this purpose.   1. The idea of the existence of purely spatial, i.e. static, regions in the Universe is worth being discussed from the modern cosmological point of view. Indeed, already the Big Bang theory, which is commonly accepted since decades, states that the Universe is expanding homogeneously, i.e. it has no non-expanding regions. For example, in 1948, a similar idea was put forward by Hoyle, Bondi, and Gold, saying that the Universe could be static despite the increasing distances between the galaxies. It was based on the assumption of a continuous creation of matter out of the vacuum. That matter was assumed to be filling out the voids which are being continuously formed between the running-away galaxies, thereby leading to the creation of new galaxies. The mechanism of the spontaneous creation of matter was remaining unclear, but the necessary rate of matter creation was found to be as low as just a few atoms per 1 cm^3 per century, so that there were no observations at that time which could disprove this phenomenon. Of course, that idea was ruled out with the development of observational data advocating in favor of the Big Bang theory, such as the qualitative difference between the most distant and the neighboring galaxies. So, I would suggest to critically review in this talk the assumption of the existence of static regions in the Universe.   2. Likewise, the hypothesis of time reversal in some regions of the Universe (cf. Ref. 12) seems to contradict the observational cosmological data. Those data rather support the steady flow of time everywhere in our Universe, starting from the moment of the Big Bang, when time emerged together with the three spatial dimensions. Apart from the cosmological aspects, it is worth clarifying how such laws of physics as conservation of energy or the second law of thermodynamics would be respected in the regions with reversed time.   3. The possibility of having a 4D Universe with two times was ruled out in the same year of 1984. Indeed, it would mean a Universe with only two spatial dimensions. Gravity in such a Universe was studied in the paper 'General relativity in a (2+1)-dimensional space-time' by J. Richard Gott III and Mark Alpert (Gen. Rel. and Grav. 16 (1984) 243). It was shown there that there is no gravitational attraction between masses in such a space, so that e.g. planetary systems cannot exist. The space in that case only gets curved at the place of the actual location of a mass, rather than everywhere, as in the physical case of three spatial dimensions.    Also, it is necessary to clarify the appearance of the same symbol 'K' in Eq. (1) and, twice, around Eq. (6).    

Author Response

(The authors gave the same response as above.)

Round 2

Reviewer 2 Report

All issues mentioned previously have been clarified.

Author Response

Dear Reviewer,

Thank you for your comments.

Sincerely yours, Tatyana Shestakova.

Reviewer 3 Report

In the revised version of the manuscript, the author has provided the necessary comments and references which clearly address points 2 and 3 from my initial report. Concerning point 1 from that report, as mentioned there, the model by Hoyle, Bondi, and Gold was assuming the staticity of the *entire* Universe (see e.g. https://en.wikipedia.org/wiki/Steady-state_model). I agree with the author that one should distinguish such a static Universe, or some imaginary static regions of our Universe, from the imaginary purely spatial regions with metric (+,+,+,+), in which time does not exist altogether. In order to avoid potential readers' confusion, I can suggest the author to include the corresponding clarifications into the revised version of the manuscript. Otherwise, the manuscript can be published in its present form, after making the necessary spelling checks.  

Author Response

Dear Reviewer,

Here is my answer to your comments:

In the revised version of the manuscript, the author has provided the necessary comments and references which clearly address points 2 and 3 from my initial report. Concerning point 1 from that report, as mentioned there, the model by Hoyle, Bondi, and Gold was assuming the staticity of the *entire* Universe (see e.g. https://en.wikipedia.org/wiki/Steady-state_model). I agree with the author that one should distinguish such a static Universe, or some imaginary static regions of our Universe, from the imaginary purely spatial regions with metric (+,+,+,+), in which time does not exist altogether. In order to avoid potential readers' confusion, I can suggest the author to include the corresponding clarifications into the revised version of the manuscript. 

I included my remarks on this point on page 2 of the revised version, in the second paragraph and added references to the works of Einstein, Hoyle, Bondi and Gold [4,5,6].

Sincerely yours, Tatyana Shestakova.